# Assessment of Hygiene Management Practices and Comparative Analysis of Regulatory Frameworks for Shared Kitchens across Different Countries

**DOI:** 10.3390/foods13060918

**Published:** 2024-03-18

**Authors:** Yu Jin Na, Jin Young Baek, So Young Gwon, Ki Sun Yoon

**Affiliations:** 1Department of Food and Nutrition, College of Human Ecology, Kyung Hee University, 26 Kyungheedae-ro, Dongdaemun-gu, Seoul 02447, Republic of Korea; yjna1@foodinfo.or.kr (Y.J.N.); jybaek@foodinfo.or.kr (J.Y.B.); 2Department of Food and Nutrition, Chung-Ang University, 4726 Seodong-daero, Daedeok-myeon, Gyeonggi-do, Anseong-si 17546, Republic of Korea; herotigger@naver.com

**Keywords:** shared kitchen, food service, regulation, hazard analysis, hygiene management

## Abstract

Shared kitchens, where users share kitchen space, are becoming popular worldwide due to the economic cost savings of startup businesses. This study conducted monitoring of microbial and chemical hazards from prepared foods and the environment of shared kitchen facilities, surveyed shared kitchen operators, and compared shared kitchen regulations between Korea and other countries. The monitoring results indicate that the hygiene status of the facilities and the microbial and chemical hazards in the prepared foods were all within the standard specifications, showing significantly lower levels compared to regular restaurants (*p* < 0.05). In particular, concurrent-use and time-division types of open shared kitchens showed significantly lower levels of both hazards than separated-individual kitchens. Survey results of hygiene inspection also confirmed better hygiene management in concurrent-use and time-division types of open shared kitchens in Korea. However, more frequent cleaning and disinfection, hygiene inspections, and training are high economic burdens in the operation of shared kitchens compared to regular restaurants. Moreover, mandatory insurance subscriptions, the operator’s responsibility in hygiene-related incidents, and high operational costs collectively challenge shared kitchens’ competitiveness in the food service market. Critical reassessments of regulations utilizing the benefits of shared kitchens are needed to promote a safe dining culture and the growth of shared kitchen startup businesses.

## 1. Introduction

The concept of shared kitchens as a food service was first proposed in the United States in the 1980s. It is a system that embodies the shared economy, focusing on cost reduction and high efficiency by sharing a space equipped with communal kitchen facilities and ownership of properties [1,2]. The advantages of shared kitchens include the cost reduction for initial startup for the food service industry and the provision of efficient production spaces [3,4]. They are seen as business incubators that help new entrepreneurs by hosting them, thereby overcoming the burdens of the early stages of a business and aiding in their independence and success [5]. In the United States, the average initial startup cost for a restaurant is USD 475,500, whereas the average initial cost for a shared kitchen focusing on takeout and delivery is USD 150,500 [6]. The market size of shared kitchens in the U.S. was USD 519.6 billion in 2020 and is expected to grow at an average rate of 12.4% annually from 2021 to 2028 [7].

The shared kitchen service “WeCook” was first introduced and implemented by a simple project company in 2015, Korea. Since 2018, there was a rapid market growth, with approximately 20 companies and about 40 stores expanding within a year or two [8]. This phenomenon is seen because of a 14.5% increase in the proportion of single-person households from 20% in 2010 to 34.5% in 2022. This increase reflects not only the shift in food culture from home-cooked meals to dining out and delivery services but also the rise of social dining. Moreover, it is the outcome of the implementation of regulatory sandboxes at the governmental level, enabling experimental practices within a regulatory-free environment for shared kitchen [9]. The growth of shared kitchens has extended beyond merely renting kitchen space to a variety of services, such as new menu development, marketing, operations, registration with smartphone delivery applications and agencies, and accounting services [8]. However, during the COVID-19 pandemic, the spread of contactless culture, restrictions on gathering sizes, and limitations on operation hours for restaurants led to the closure of many shared kitchens in Korea. Even with the transition to an endemic phase of COVID-19 in 2023, the recovery of reduced market of shared kitchens remains difficult in Korea.

The downsizing of the shared kitchen market can be attributed to several causes, but the most significant factor is attributed to policy regarding shared kitchens. Previously, Korea’s Food Sanitation Act was established to allow only one operator in a kitchen, due to concerns over cross-contamination and following food poisoning [10]. To facilitate the shared kitchen business, a regulatory exception was also introduced, allowing temporary permits for multiple operators to conduct business in a single kitchen [8]. Shared kitchens, being utilized by multiple individuals for food preparation, are especially prone to cross-contamination [11]. Microbiological analysis conducted on kitchen sponges, draining racks, and sink drains in shared kitchens unveiled that the kitchen sponge exhibited the highest levels of bacterial contamination, followed by the draining rack and then the sink drain. This highlights the need for enhanced hygiene measures to mitigate the risk of cross-contamination in shared kitchens [12]. This risk is heightened when ready-to-eat products, such as salads, are produced in these communal spaces.

However, the responsibility for sanitation issues like food poisoning in shared kitchens falls on the single registered operator, despite the shared business model of these kitchens [10,13]. Therefore, the shared kitchen market is experiencing a downsizing in Korea due to the policy of a significant economic burden on the principal operator, contrary to the global growth trend of shared kitchens.

Consequently, this study was conducted (1) to investigate the current operational conditions of various types of shared kitchens in Korea. This involved visiting various shared kitchens across the country, inspecting their operational conditions, and surveying operators to gather information on their operational status, (2) to assess the potential biological and chemical hazards associated with both the prepared foods and the environment within shared kitchen facilities, and (3) to draw insights from shared kitchen models implemented overseas to optimize the advantages of the shared kitchen system in Korea.

## 2. Materials and Methods

### 2.1. Classification and Status of Shared Kitchen Models

To understand the current state of the domestic shared kitchen industry, we visited 30 shared kitchen companies nationwide and classified their business models, as shown in Figure 1.

### 2.2. Status of Sanitation Management in Shared Kitchens Facilities

Among the 30 shared kitchen companies, 11 facilities were selected for analysis of facility sanitation management. To monitor microbial contamination of environment, swab kits (3M Pipette Swab, 3M, Saint Paul, MN, USA) were used on the hallway from the kitchen to the exit leading to the outside, the door knob, inside of refrigerators, sinks and worktops, cutlery (knife, cutting board, mixing bowls, etc.), and cooking clothes for airborne bacteria, total bacterial count, and coliforms.

### 2.3. Quantitative Microbiological Analysis of Prepared Foods in Shared Kitchens

Seven products, including frozen breads (cheese cake, plain scone), snacks (peanut butter with nuts and chocolate cookies), ready-to-eat (RTE) salad (quinoa tofu mushroom, shrimp quinoa), and sauces (yuja dressing dressing), were selected from 30 shared kitchen companies across the country to evaluate microbial and chemical hazards of prepared foods in shared kitchens. For hygiene level, quantitative analysis of total bacterial counts, coliforms, and *E. coli* was conducted in all samples according to the Food Code [13]. Twenty-five grams of the sample were homogenized for 2 min with 225 mL of sterilized 0.1% peptone water (BD, Sparks, MD, USA) in a sterile bag (3M 1020W bag, 3M, Saint Paul, MN, USA). One milliliter of the homogenized solution was diluted in 9 mL of sterilized 0.1% peptone water, and 1 mL of the diluted solution was placed onto 3M Petri films (AC 3M Petrifilm^TM^, 3M, Saint Paul, MN, USA), incubated at 36 °C for 48 h, and then we counted the number of red colonies for total aerobic bacteria. For coliforms and *E. coli*, 1 mL of diluted solution was placed onto *E. coli*/coliform Petri films (EC/CC 3M Petrifilm^TM^ 3M, Saint Paul, MN, USA), incubated at 37 °C for 48 h, and then we counted the number of red and blue colonies with bubbles.

Quantitative and qualitative analyses of pathogenic food poisoning bacteria were conducted on all samples [13]. For quantitative analysis of *Staphylococcus aureus, Bacillus cereus*, and *Vibrio parahaemolyticus*, 25 g of the sample was homogenized with 225 mL of sterilized 0.1% peptone water in a sterile bag for 2 min. The homogenized solution was then diluted with 9 mL of sterilized 0.1% peptone water. One milliliter of the diluted solution was spread onto Baird–Parker agar (BPA, Oxoid, Hampshire, England) for *S. aureus* and incubated at 36 °C for 48 h, as well as onto Mannitol–Yolk–Polymyxin (MYP, Oxoid, Hampshire, UK) B agar for *B. cereus*, and incubated at 30 °C for 24 h. One milliliter of the diluted solution was spread onto thiosulfate citrate bile salt sucrose (TCBS agar, Oxoid, Hampshire, UK) agar for *V. parahaemolyticus*, and incubated at 36 °C for 24 h. The suspect colonies were all counted.

For the qualitative analysis of *Listeria monocytogenes*, 25 g of the sample was incubated with 225 mL of sterilized PALCAM broth (Oxoid, Hampshire, UK) at 30 °C for 48 h. Then, 0.1 mL of the culture was inoculated into Fraser broth and further incubated at 36 °C for 24 h for secondary enrichment. One mL of the enriched culture was then spread onto PALCAM agar and incubated at 36 °C for 48 h. For the qualitative analysis of Salmonella, 25 g of the sample was homogenized with 225 mL of sterilized 0.1% peptone water in a sterile bag for 2 min, then incubated at 36 °C for 24 h. Afterward, 1 mL of the culture was added to 10 mL of tetrathionate broth (TT broth, Oxoid, Hampshire, UK) and incubated at 36 °C for 24 h for enrichment. Simultaneously, 0.1 mL of the culture was added to 10 mL of Rappaport–Vassiliadis broth (RV broth, Oxoid, Hampshire, UK) and incubated at 41.5 °C for 24 h for enrichment. Then, 1 mL of the enriched culture was spread onto xylose lysine deoxycholate (XLD agar, Oxoid, Hampshire, UK) and incubated at 36 °C for 24 h to check for the presence of suspect colonies.

### 2.4. Analysis of Chemical Hazards in the Prepared Foods in Shared Kitchens

Samples were tested for total aflatoxins (peanut butter cookies), residual pesticides (RTE salads), preservatives (bread, sauce), unapproved tar colors (sauce), and heavy metals (RTE salads) according to the standards in the Food Code [14]. For total aflatoxin analysis, 25 g each of peanut butter chip cookies were homogenized in 100 mL of 70% methanol solution for 5 min and filtered through Whatman No. 4 paper (Whatman Co, LTD., Maidstone, UK). Then, 10 mL of filtrate was mixed with 30 mL of 1% Tween 20 (Sigma-Aldrich Co, LTD., St. Louis, MO, USA) in a 100 mL flask and filtered using a glass fiber filter (AP4002500, Sigma-Aldrich Co, LTD., St. Louis, MO, USA). Then, 20 mL of the extract was passed through an immunoaffinity column (AflaStar^TM^ R, Romer Labs Co, LTD., Getzersdorf, Austria) at a rate of 1 drop per second. For secondary extraction, 10 mL of distilled water and 3 mL of acetonitrile (HPLC grade, Daejung Co, LTD., Siheung, Kyeonggi-do, Republic of Korea) were passed in order through at the same velocity. Then, 0.8 mL of a 20:80 (*v*/*v*) mixture of acetonitrile and water (HPLC grade) was added and filtered through a 0.25 μm syringe membrane filter (SRP25, SIBATA Co, LTD., Tokyo, Japan) for the test solution. Total aflatoxins (B1, B2, G1, G2) were analyzed using a high-performance liquid chromatograph (HPLC, LC-2030C NT, SHIMADZU, Kyoto, Japan). The analysis used a Kromasil C18 HPLC column (4.6 mm × 250 mm, 5 μm, Bohus, Sweden), with a mobile phase of acetonitrile and water at a 25:75 (*v*/*v*) ratio. The excitation wavelength was set at 360 nm, and the fluorescence wavelength at 450 nm, with a flow rate of 1 mL/min and an injection volume of 10 μL. The areas for aflatoxins B1, B2, G1, and G2 were quantified and total aflatoxin was calculated.

To verify the compliance with residual pesticide standards for quinoa tofu mushroom and shrimp quinoa salads, a multiresidue method for 320 types of pesticides was detected using HPLC. Each 30 g sample was vigorously shaken for 2 min with 350 mL of 70% acetone and then filtered using Whatman No. 4 paper (Whatman Co, LTD., Maidstone, UK). Then, 80 mL of the filtered solution was transferred to a separating funnel, and 100 mL of petroleum ether was added and shaken vigorously for 1 min before separating into two layers. The lower layer was transferred to another separating funnel, and the upper layer was moved to a round-bottomed flask for use in a concentrator. Seven grams of sodium chloride was added to the funnel containing the lower layer and vortexed for 30 s (VM1, LABTron, Seoul, Republic of Korea), followed by the addition of 100 mL of dichloromethane, and then it was all shaken vigorously for 1 min to separate the layers. The lower layer was collected and combined with the solution in the round-bottomed flask, and concentrated using a rotary evaporator (WEV-1001V, SH scientific Co, LTD., Portland, OR, USA) at 60 °C until the final volume was 2 mL. Then, 100 mL of petroleum ether was added to the final solution and concentrated again at 60 °C until it reached a final volume of 2 mL, followed by the addition of 5 mL of acetone and filtration through a 0.25 μm syringe membrane filter to prepare the final test solution. For HPLC analysis, a μ-Bondapak C18 column (4.6 mm × 300 mm, 10 μm, Waters^TM^ Co, LTD., Milford, MA, USA) was used with an HPLC-UVD detector, and 1 L of water was mixed with 35 mL of phosphoric acid at a flow rate of 1.0 mL/min and an injection volume of 10 μL.

To determine the compliance of preservative standards in yuja dressing as a sauce, the contents of sorbic acid and parahydroxybenzoic acid were analyzed using HPLC. Five grams of yuja dressing were mixed with 30 mL of ethanol in a 50 mL volumetric flask. To this, 2 mL of 15% potassium ferrocyanide solution was added and mixed, followed by the addition of 2 mL of 30% zinc sulfate solution, and shaken for 2 min. The prepared solution was treated in a sonicator (Powersonic 520, Hwasjom tech Co, LTD., Dae-gu, Republic of Korea) for 30 min, then ethanol was added up to the mark. After centrifugation at 4000 rpm for 10 min (Z 307, Hermle, Gosheim, Germany), the supernatant was collected and filtered through a 0.25 μm syringe membrane filter (SRP25, SIBATA Co, LTD., Tokyo, Japan) to prepare the test solution. The HPLC analysis used a Capcell pak MF-C8 column (4.6 mm × 150 mm, 4.5 μm, Osaka soda Co, LTD., Osaka, Japan), and detection was carried out at 235 nm using a UV detector. The flow rate used a gradient method with mobile phase A consisting of a solution diluted in 1 L of distilled water containing 2.5 g of 40% tet-rabutylammonium hydroxide and 1.2 g of 85% phosphoric acid, and mobile phase B was acetonitrile. The gradient conditions for both solvents (A and B) are shown in Table 1. The amount of preservatives was quantified using the areas of the detected peaks.

Tar colors in yuja dressing were analyzed to determine compliance with standard specifications. Briefly, 5 mL of the sample was taken into a 100 mL beaker, followed by the addition of 50 mL of distilled water and stirred for 10 min. Then, 5 mL of this solution was then adjusted to pH 3–4 with 10% acetic acid. A 0.1 g white cotton ball was added and stirred for 30 min to observe dying of the wool for detection.

To verify compliance with the standards for heavy metals, specifically lead and cadmium, 10 g of quinoa tofu mushroom and shrimp quinoa salads were placed on a platinum dish and carbonized at 300 °C for 10 min, followed by ashing at 450 °C for 2 h in a 1500 °C Muffle furnace (SH scientific Co, LTD., Portland, OR, USA). The ashed residue was treated with 20 mL of 1% nitric acid to prepare the test solution, which was analyzed for cadmium and lead content using ICP-OES (GBC Integra-XMP, Braeside, Australia).

### 2.5. Assessment of Food Hygiene Compliance in Shared Kitchen Companies

For the survey study, 30 shared kitchen companies were visited nationwide, and their compliances for food hygiene were assessed using a total of 128 evaluation items. The evaluation categories included compliance with regulations (18 items), workstation and environment (47 items), hygiene management (51 items), food defense (2 items), and document management (10 items). The evaluators were three professionals from food hygiene consulting firms in Korea. Each item was scored 0 for compliance and 1 for noncompliance.

### 2.6. Survey on Operational Management Practices and Regulatory Awareness among Managers in the Shared Kitchen Industry

A survey study was conducted with 20 operators and hygiene managers to investigate methods of hygiene management and perception of regulations of shared kitchen industry. The questionnaire was divided into basic information about the establishment (14 items), facility and hygiene management (25 items), and regulatory improvements (7 items).

### 2.7. Comparison of the International Shared Kitchen Systems

To compare the status, regulations, and application of HACCP in shared kitchens abroad, we conducted a literature review analyzing published papers, national policy reports, and related laws. Particularly, our examination focuses on the United States and Europe, where the shared kitchen concept emerged relatively early. In the United States, we analyzed operational regulations in major cities with active shared kitchen scenes, such as New York City, Chicago, and Georgia, using data provided by “The Food Corridor”, a brokerage firm offering information on shared kitchens. Our analysis then extends to Europe, with a focus on the EU, the UK, and France. Additionally, our research highlights China and India in Asia, where the shared kitchen market has notably expanded. Our comparative analysis across these countries primarily centers on the terminology used for shared kitchens, the roles of hygiene managers, and the compulsory regulations regarding liability insurance that are established domestically.

### 2.8. Statistical Analysis

To determine the statistical significance of each item, an analysis was conducted using one-way analysis of variance (ANOVA) in PASW Statistics 18 software (SPSS Inc., Chicago, IL, USA), followed by post hoc analysis to determine which groups are significantly different using Duncan’s multiple range test (*p* < 0.05).

## 3. Results and Discussion

### 3.1. Classification and Operational Conditions of Domestic Shared Kitchen Models

The shared kitchen industry was generally classified into open-type, shared kitchen and separated-type shared kitchens. Open-type shared kitchens were further divided into concurrent-use and time-division types, in which multiple businesses use the kitchen at the same or different times. On the other hand, the separated-type shared kitchens use individual kitchens for each business but share storage facilities and hallways (Figure 1).

Among 30 nationwide shared kitchens, the separated-type shared kitchen was the most common (16), followed by the time-division (9) and concurrent-use types (5) of open-type shared kitchens. Each type had slightly different business models; the separated-type shared kitchens mainly had a food court business model in a restaurant or rest area and were essentially exempted from shared kitchen regulations. The time-division types were mostly found in domestic highway rest areas, selling foods, snacks, etc., and the business operators use the same facility at different times. Moreover, the typical business model of shared kitchens is the concurrent-use type, which includes cooking classes, entrepreneur training, and government-supported projects, with one-time or regular reservation users. Both the concurrent-use and time-division types of shared kitchens are currently subjected to the Food Sanitation Act’s regulations, requiring mandatory selection of a hygiene manager, completion of hygiene training at least once a year, and insurance enrolment. In the case of hygiene management issues, the hygiene manager must take responsibility for all related documentation and duties [10,15].

### 3.2. Status of Sanitation Management in Shared Kitchens Facilities

The contamination levels of hygiene-indicator bacteria and food poisoning pathogens in 11 domestic shared kitchen facilities are presented in Table 2. Regardless of the type of shared kitchen, *S. aureus*, *B. cereus*, *Salmonella* spp., and yeast and mold were not detected in any environment. However, very low levels of total aerobic bacteria and coliforms were detected in all environments except hallways, and the degree of contamination levels varied according to the type of shared kitchen. In all environments, the lowest levels of total aerobic bacteria and coliforms were detected in concurrent-use type shared kitchens.

The levels of total aerobic bacteria and coliforms within the refrigerator and cutlery varied significantly based on the type of shared kitchen (*p* < 0.05). The total aerobic bacteria inside the refrigerator were 1.64 ± 1.68, 2.31 ± 2.22, and 3.90 ± 2.44 log CFU/100 cm^2^ for concurrent-use, time-division, and separated-individual type of shared kitchen, respectively. The total aerobic bacteria inside the refrigerator was significantly low in concurrent-use type (*p* < 0.05). The lowest level of coliforms inside the refrigerator was observed in time-division (0.12 ± 0.29 log CFU/100 cm^2^, followed by concurrent-use type and separated-individual type (*p* < 0.05). For cutlery, there was no statistically significant difference in coliforms, but a significant difference in total aerobic bacteria contamination levels was observed among the types of shared kitchens. The total aerobic bacteria count was notably highest in the separated-individual type at 3.57 ± 2.21 log CFU/100 cm^2^, followed by the time-division method (2.16 ± 1.80) and the concurrent-use method (0.88 ± 1.42) (*p* < 0.05). Although not statistically significant (*p* > 0.05), the highest contamination levels of total aerobic bacteria (4.48 ± 1.22) and coliforms (2.54 ± 3.59) were the sink and worktop in the time-division type, followed by the separated-individual type.

Previous research [16] showed that the average total aerobic bacteria and coliforms of knives was 2.47 × 10^6^ CFU/100 cm^2^ and 0.51 × 10^5^ CFU/100 cm^2^, respectively, in Korean restaurants. The average levels of total aerobic bacteria and coliforms were also very high, at 2.79 × 10^6^ CFU/100 cm^2^ and 0.63 × 10^5^ CFU/100 cm^2^, respectively, in the interiors of refrigerators. Moreover, in 10 Japanese restaurants recognized for their exemplary hygiene practices, the contamination levels of total aerobic bacteria, coliforms, and *S. aureus* on cutting boards were measured at 2.25 ± 0.96 log CFU/100 cm^2^, 2.22 ± 0.53 log CFU/100 cm^2^, and 0.30 ± 0.48 log CFU/100 cm^2^, respectively [17]. Comparing the results of this study with previous research on the microbial contamination levels in shared kitchens and regular restaurants, the microbial contamination levels of facilities in shared kitchens, particularly concurrent-use and time-division types, are lower than those in regular restaurants. This indicates that the overall hygiene standards in shared kitchen management, irrespective of the type, are notably superior to those observed in typical restaurants across Korea. This distinction can be attributed to the implementation of more rigorous regulations related to shared kitchens, such as the appointment of hygiene managers and mandatory hygiene training sessions conducted at least annually [10]. These measures are notably more stringent compared to the requirements imposed on conventional restaurants.

#### 3.2.1. Microbial Quantitative Analysis of Prepared Foods in Shared Kitchens

The contamination levels of hygiene-indicator bacteria and food poisoning pathogens of prepared food in shared kitchens are presented in Table 3. No food poisoning pathogens were detected, and coliforms and *E. coli* were also not detected, except for in two kinds of quinoa salads. The contamination levels of hygiene-indicator bacteria were significantly highest in salads (*p* < 0.05), whereas other products such as bread and snack items exhibited very low microbial contamination levels, typically less than 1 log CFU/g. The quinoa tofu mushroom salad had total aerobic bacteria and coliforms at levels of 3.93 ± 0.24 log CFU/g and 1.42 ± 0.15 log CFU/g, respectively, while the shrimp quinoa salad had 3.27 ± 0.68 log CFU/g and 0.97 ± 0.60 log CFU/g, respectively. It is believed that RTE products like salads exhibit higher levels of microbial contamination compared to other product categories because they do not undergo any special food processing methods or use additives, except for washing [18,19].

The contamination levels of total aerobic bacteria, coliforms, and yeast and mold in fresh-cut fruit cups sold in Korea were measured at 3.10 ± 1.14 log CFU/g, 1.28 ± 0.96 log CFU/g, and 3.71 ± 0.81 log CFU/g, respectively. For fresh-cut vegetables, the contamination levels were recorded at 4.32 ± 1.16 log CFU/g for total aerobic bacteria, 1.04 ± 0.75 log CFU/g for coliforms, and 2.78 ± 0.97 log CFU/g for yeast and mold [20]. Another study revealed that the contamination levels of total aerobic bacteria and coliforms in lettuce salads sold in markets were 4.24 ± 0.01 log CFU/g and 1.08 ± 0.01 log CFU/g, respectively [17]. In this work, the contamination levels of total aerobic bacterial of two kinds of quinoa salads prepared in shared kitchens were about 1.0 log CFU/g, which is lower than those of vegetable salads sold in convenience stores and markets.

#### 3.2.2. Chemical Hazard Analysis of Foods Prepared in Shared Kitchens

The analysis results of chemical hazard including pesticide residue and heavy metals in seven prepared foods are presented in Table 4. No preservatives in the cheesecake and plane scone and total aflatoxin in peanut butter cookies were detected (data not shown). However, some pesticide residues and heavy metals were detected in the two kinds of quinoa salads. According to the analysis results of 320 kinds of pesticides, only dinotefuran, imazalil, lufenuron, and pyridalyl were detected in quinoa salads. Among pesticide residues, the shrimp quinoa salad contained the significantly highest levels of imazalil at 0.068 mg/kg, while the quinoa tofu mushroom salad had the highest levels of pyridalyl at 0.054 mg/kg (*p* < 0.01). The level of dinotefuran in the quinoa tofu mushroom salad was approximately twice that found in the shrimp quinoa salad, and the level of imazalil was about 2.5 times higher in the shrimp quinoa salad than the quinoa tofu mushroom salad. This is likely due to the addition of eggs, kidney beans, and corn in the shrimp quinoa salad. Imazalil is primarily used for presowing treatment of barley and wheat seeds and for disinfecting egg hatchery equipment before egg collection [21]. In previous work, chlorfenapyr and lufenuron, as pesticide residues, were detected in salad vegetables, each at 5.0 mg/kg [22], and lead was also detected at a level of 0.38 ± 0.05 mg/kg [23]. Comparing these previous studies with the results of this research, lufenuron was commonly detected in salads, but the amount of lufenuron in salads sold in shared kitchens was about 1/200th of the level of previous work. Lufenuron is commonly detected as a residue in agricultural products, with permissible limits set at 1.0 mg/kg (ppm) for oranges, 0.3 mg/kg for apples, 0.2 mg/kg for cucumbers, 5.0 mg/kg for lettuce, and 0.5 mg/kg for cabbage. Ready-to-eat foods like salads are allowed to contain residues within the permissible limits established for the raw food ingredients [24].

In the quinoa tofu mushroom salad, a significantly higher level of cadmium was detected at 0.0095 mg/kg compared to the shrimp salad, which contained 0.0033 mg/kg (*p* < 0.05). Additionally, lead was detected at 0.0042 mg/kg in the quinoa tofu mushroom salad and 0.0047 mg/kg in the shrimp quinoa salad (Table 4). Quinoa tofu mushroom salad showed three times higher levels of cadmium than shrimp quinoa salad, but no significant difference between two kinds of salad was observed in the lead level (*p* > 0.05). Lead was also found at lower levels in quinoa salads in shared kitchens than those found in quinoa salads in general supermarkets and convenience stores. According to the results of previous studies, dimethomorph in quinoa was detected at the average of 3.33 ± 0.25 mg/kg [25], and heavy metals such as aluminum, arsenic, and cadmium were detected at levels of 11.5 ± 6.5, 0.028 ± 0.05, and 0.049 ± 0.03 mg/kg, respectively, while mercury was not detected [26].

The results of this study indicate that salad products manufactured in shared kitchens seem to be handled with greater care regarding chemical hazards compared to products available in traditional supermarkets and convenience stores. Commercially produced salads, which undergo bulk washing and distribution, may not effectively eliminate pesticide residues to the same degree as those produced on a smaller scale in shared kitchens.

Small-scale multiple washing processes are conducted to prepare salads in shared kitchens. Pesticide residues on leafy vegetables show an average reduction over 77% when washed with running water [27,28]. Simple washing multiple times with running water can effectively remove pesticide residues [29].

### 3.3. Survey Study for Hygiene Conditions of Shared Kitchen

A hygiene inspection survey was conducted by visiting 30 shared kitchen businesses across the country, and the results are presented in Table 5. In the survey study, a score of 0 was assigned for compliance and 1 for noncompliance in each item, indicating that a high average score represents poor management. Although no significant difference was observed among the types of shared kitchens, the concurrent-use type scored lower than the other two types in the “workstation and environment” and “hygiene management” categories. In the score of “legal compliance” category, significant differences among all three types (*p* < 0.05) were found. The highest score was the separated-individual type (2.81 ± 1.72), followed by the time-division type (2.11 ± 2.09) and the concurrent-use type (0.60 ± 0.55), indicating that better management of shared kitchen regulation is observed in the concurrent-use and time-division types. Similarly, the concurrent-use and time-division types scored significantly (*p* < 0.05) lower, with 2.20 ± 2.05 and 1.89 ± 2.71, respectively, in the “document management” category, compared to the score (5.25 ± 2.74) of separated-individual type. The Food Sanitation Act mandates the appointment of a hygiene manager in shared kitchens and includes document management in their responsibilities [10]. The results of the survey study showed that document management was less fulfilled in separated-individual type, which is the unregulated type of shared kitchen. The “food defense” category is primarily to control the external visitors.

Given the shared kitchen’s nature of multiple users sharing a space, control of external visitors is vital for traceability in case of hygiene issues. The concurrent-use type, having the highest foot traffic as a business model, recorded significantly (*p* < 0.05) lower scores in food defense compared to the time-division and separated-individual types. This can be due to the implementation of visitor logbooks and separate areas for delivery personnel in the concurrent-use type. Overall, the hygiene inspection results indicated that hygiene management was most effectively implemented in the concurrent-use type.

### 3.4. Survey Study of Operation Management in Shared Kitchens

A total of 51 questions related to operational management were surveyed with 20 operators of shared kitchens in Korea (2 concurrent-use type, 4 time-division type, 14 separated-individual type), and the most significant and meaningful results were selected and are presented in Table 6. Participants in the survey study were divided into two categories: shared kitchen operators (CEOs), comprising 45% (9 out of 20), and safety managers, constituting 55% (11 out of 20).

For the question of accomplishment and frequency of cleaning and disinfection (Q28), 35% and 20% of the respondents cleaned and disinfected once a month and once a week, respectively. Twenty percent of the respondents reported that they cleaned and disinfected the facility daily. Especially, operators in open-shared kitchens showed a high level of hygiene management due to their responsibility of shared kitchen regulations. On the other hand, operators of the nonregulated, separated-individual type appeared to have lower frequency and burden of cleaning and disinfection.

In response to hygiene inspections frequency of the shared kitchen (Q31), 35% (7/20) of all shared kitchen operators inspected once a month, followed by once a week (25%. 5/20). Fifty percent of separated-individual type kitchens reported that they conduct a hygiene inspection once a month, suggesting that hygiene management is more rigorously conducted in concurrent-use and time-division types of open shared kitchens.

In response to hygiene training frequency by the hygiene manager (Q35), 35% of the operators of all shared kitchens (7/20) had training once a month. Both concurrent-use and time-division types had a more frequent hygiene training schedule. In contrast, 64.3% of separated-individual type kitchens had no internal hygiene training, followed by once a month and once a year at 28.6% and 7.1%, respectively. Under the regulation in Korea, operators and hygiene managers in shared kitchens are mandated to conduct at least one hygiene training session per year [10]. The high rate of no internal hygiene training was noticeable in the separated-individual type. Additionally, practical hygiene training is conducted more frequently in the concurrent-use and time-division types than the separated- individual type. Since the operator takes full responsibility in open-shared kitchens, it seems that practical hygiene education is conducted more frequently.

Lastly, the major challenges in the shared kitchen (Q43) were surveyed to determine the most difficult aspects of operation. The economic cost for operation received the highest response, followed by hygiene management of shared kitchen users. Operators of shared kitchens, regardless of operation types, reported that operational economic costs, including mandatory insurance, frequent hygiene training, inspections, and cleaning and disinfection are the major challenges to manage the shared kitchen, compared to regular restaurants. Although shared kitchens initially gained attention for reducing startup costs through business incubation and kitchen sharing, the current regulations increase operational costs, thereby reducing the market competitiveness of shared kitchens in Korea.

### 3.5. Comparison of Shared Kitchens Regulations Abroad

In the early days, shared kitchens were denoted by a range of terms, including shared kitchen, cloud kitchen, and community kitchen. Recently, there has been a trend to distinguish between shared kitchens and cloud kitchens. In particular, cloud kitchens, also known as “dark kitchens” or “ghost kitchens”, primarily refer to kitchens specialized in delivery without physical dining spaces, evolving from the concept of shared kitchens. Table 7 compares the terminology, regulations, HACCP application, and insurance of shared kitchens in different countries.

#### 3.5.1. United States of America (USA)

The number of shared kitchen businesses has been continuously increasing, from 200 in 39 states in 2016 to 600 in 48 states by 2020 [30,31,32]. Food service facilities are regulated by the local governments in each state, and food service businesses are managed by the local health departments [30]. The key laws and regulations of shared kitchens in New York State and City of Chicago were compared in the USA [33,34].

In New York State, shared kitchens operators and users must follow the laws of the NYC Health Code [35] to obtain a permit to operate a Food Service Establishment (FSE) beforehand. The NYC Health Code then differentiates and regulates both operators and users of shared kitchens. The NYC Department of Health and Mental Hygiene mandates an agreement between operators and users for hygiene management, specifying the purpose of the shared kitchen, types of food to be sold, and places of sale. The regulations enforce strict responsibilities on each business owner, thereby increasing accessibility to the sector and encouraging efficient resource utilization.

In Chicago, shared kitchens are governed by the Chicago Department of Public Health’s Food Safety Regulation, which includes food sales permits, commercial tax charges, safe food manufacturing and storage methods, and hygienic environments [36]. To reduce the initial operational burden, Chicago offers the Community Kitchen Program, providing simplified licensing for operators and users, food safety management training for operators, and various equipment and facilities necessary for starting a shared kitchen. Similar to New York State, both operators and users in Chicago must have their licenses and have responsibility for hygiene issues [37]. However, Chicago has an additional distinction: the licenses required by shared kitchen operators are differentiated based on whether the end consumer is an individual or a corporation. Chicago’s Community Kitchen Program generally reduces the operators’ burden by offering initial cost reductions and safety training through government support, donations from nonprofit organizations, and community participation. Additionally, both operators and users share responsibility of hygiene issues in shared kitchen.

Likewise, Georgia has established rigorous food safety regulations [38], overseen by the Georgia Department of Agriculture, which meticulously monitors food processing plants operating as food sales facilities. Additionally, the department provides comprehensive guidelines for shared kitchens, clearly delineating the responsibilities of managers and operators [39]. This framework ensures that operators oversee users effectively, consolidating safety management within shared kitchen environments.

On the other hand, operators in shared kitchens all have responsibility for hygiene issues in Korea. Moreover, shared kitchen operators face high economic costs due to requirements like employing sanitation managers and mandatory insurance.

#### 3.5.2. Europe

In Europe, shared kitchens are referred to as “Community/Shared Kitchens” and are regulated based on the EU’s General Food Law [40]. In the UK, such commercial kitchen spaces are often referred to as “commissary kitchens”, “dark kitchens”, “virtual kitchens”, or “ghost kitchens”. These terms are particularly used to describe kitchens that specialize in preparing food solely for delivery, with customers placing orders online rather than dining in. In Europe, food business operators, including those in shared kitchens, must maintain a food safety management system, possibly with HACCP-based plans. Though not obligatory, having staff capable of executing food safety plans effectively is crucial. The detailed safety management laws for shared kitchens vary according to the laws of each EU member state. The shared kitchen safety regulations of the United Kingdom and France were examined as representative examples in this work.

In the UK, the management of shared kitchens falls under the UK Health Security Agency, which handles regulations, approvals and permits, and health and hygiene regulations, on a regional basis [41,42]. The approvals and permits for shared kitchens are generally under the jurisdiction of the city, district, or local government of the area, and these authorities are responsible for examining and managing various aspects of shared kitchens such as construction, hygiene, and safety [43]. Approval criteria of shared kitchens include building standards and safety regulations that cover structural safety, fire safety, entrances, emergency exits, and intended use of the building. Regulations also include the management of food processing, storage, cooking, waste disposal, noise management, ventilation systems, and disability standards, which cover physical accessibility, pathways, facilities, and services for people with disabilities [44].

The UK and France in the EU have adopted HACCP as part of their safety management systems. In UK, HACCP is implemented in shared kitchens for hazard analysis, risk control, and food safety procedures. Employees are required to undergo mandatory training in food hygiene [45]. The labeling of food and allergy information for consumer safety is also mandatory. Shared kitchen operators can have various insurance products such as public liability insurance, product liability insurance, property insurance, and accident and illness insurance [46]. Public liability insurance covers legal liability for accidents causing injury or property loss to others, and safety accidents related to food manufactured and sold in shared kitchens. Property insurance covers damage, theft, or fire to the shared kitchen building and equipment. Accident and illness insurance protects employees from economic costs due to accidents or illnesses.

In France, HACCP is mandatory for all food industries, including shared kitchens, to identify and manage risks [47]. To apply HACCP, appropriate facilities and equipment for food safety and hygiene in shared kitchens are required, along with hygiene management standards between operators and users [48]. Additionally, operators, users, and employees in shared kitchens must complete food safety training and undergo health examinations before starting work [48]. France also offers liability insurance products to reduce potential risks in shared kitchens. Business liability insurance protects against damages arising from services related to food provided by shared kitchen operators. Property insurance covers loss or damage to equipment, tools, and other assets owned by the operator. Employee liability insurance protects against accidents or damages caused by employees. Lastly, product liability insurance covers damages arising from products manufactured and sold by shared kitchen operators, including defects in food quality or food safety issues [49,50].

Comparing the EU’s shared kitchen management regulations, HACCP is not a mandatory application for shared kitchens in Korea. However, if a product produced in shared kitchen falls under the category requiring HACCP certification, it must be certified. Additionally, liability insurance in the EU is merely a recommended insurance product, not a mandatory requirement like in Korea.

#### 3.5.3. Asia

The shared kitchen market has not been developed in Asia. Notably, China has recently shown significant growth and possesses a unique form of shared kitchen business. Unlike the typical shared kitchen model, where kitchen space is shared, the shared kitchen is operated by sharing a warehouse for food supplies [5]. This business model focuses on reducing the storage space for ingredients at individual businesses and lowering operational costs through collective purchasing of ingredients. Shared kitchen platforms in China share business- and customer-related data. Thus, China has established regulations for information security and protection, as shared kitchen services involve the sharing and big data processing of customer-related information [51]. In China, regulations for shared kitchens are relatively lenient, with food safety being autonomously managed by the businesses under self-responsibility guidelines [52]. The government is simplifying registration procedures for shared kitchens, facilitating communal purchasing of ingredients, and providing big data related to dining consumption, thereby acting as an incubator for the dining business and rapidly spreading in the Chinese market [53]. Shared kitchens in China tend to adopt a tendency towards simplification, mass production, and standardization of menu offerings and operational models [54]. Regulations are also less strict in China, allowing for rapid expansion and efficient operation of shared kitchens within the country.

India sets standards and guidelines for the food industry, issuing certifications through the Food Safety and Standards Authority of India (FSSAI) license [55]. The Department of Industry and Delhi’s Dialogue and Development Commission have also revealed plans to include support for “cloud kitchen” policies in the government budget. This initiative aims to boost the food and beverage industry, highlighting a proactive approach to encouraging shared kitchen enterprises.

In Korea, shared kitchens provide customized services tailored to individual needs and require a high level of responsibility from both operators and users. There is no regulation for preapproval of operators and users, but the excessive regulation at the level of HACCP prerequisite conditions has been introduced to ensure consumer safety. It is also required to employ experienced food sanitation managers [56]. Shared kitchens, also known as kitchen incubators, are leveraging technology to offer cost-effective facilities and operating models to support food entrepreneurs. However, the excessive responsibilities placed on operators along with fixed expenses become significant burdens for individual business owners, hindering the efficient operation of shared kitchens and the revitalization of the food industry. The recent amendment to the Food Sanitation Act, specifically pertaining to shared kitchen operations, has implemented an exclusion for the separated-individual type from the shared kitchen business category. This decision was prompted by the considerable number of operators within such types and their heightened vulnerability to hygiene issues. Based on the results of this study, it is recommended that food safety responsibilities of shared kitchen operators and the mandatory insurance of shared kitchens must be reevaluated to revitalize the shared kitchen market in Korea.

**Table 7 foods-13-00918-t007:** Comparison of shared kitchen regulations abroad.

	USA			EU	UK	France	China	India	Korea
	New York	Chicago	Georgia						
Law or Regulation	New York Health Code	Municipal Code of Chicago & Chicago City of Rules	The Georgia Food Act	General Food Law	Food Safety Act 1990 & The Food Safety and Hygiene Regulations	Compliance with European Union Regulations	Food Safety Law of the People’s Republic of China	Food Safety and Standards Authority of India license	Korean Food Sanitation Act.
Official name of Shared Kitchen	Shared or Communal Kitchen	Shared Kitchen	Community Kitchen	Community or Shared Kitchen	Shared, Dark, Commissary Kitchen	Shared Food Production Facility	Shared Kitchen	Cloud Kitchen	Shared Kitchen
Hygiene manager	At least one staff member with a food protection certificate	Must employ a certified food protection manager.	One or more employees to obtain food safety manager certificate	Nonmandatory	Nonmandatory	Nonmandatory	Nonmandatory	Nonmandatory	Mandatory
HACCP application	Nonmandatory	Nonmandatory	Nonmandatory	For self inspection systems	For food safety management systems	For self-inspection systems	Recommend	Recommend	Mandatory for specific food sectors
Insurance	Nonmandatory	Nonmandatory	Nonmandatory	Nonmandatory	Nonmandatory	Nonmandatory	Nonmandatory	Nonmandatory	Mandatory
References	[35]	[36]	[38]	[40]	[41]	[40]	[52]	[55]	[56]

## 4. Conclusions

The concurrent-use and time-division types of open shared kitchens showed significantly lower levels of microbial and chemical hazard compared to the separated-individual type kitchen. Additionally, survey results of hygiene inspection revealed that scores of hygiene management, workplace environment, and regulatory compliance were significantly lower in the concurrent-use type of open-shared kitchen compared to the separated-individual type kitchen, confirming better hygiene management in open shared kitchens. Survey results of operators revealed that frequent cleaning and disinfection of facilities, shorter hygiene inspection cycles, and regular implementation of hygiene training also become economic burdens on shared kitchen operations. The results of regulation comparison of shared kitchens abroad also suggest that food safety responsibilities of both operators and users must equally imposed. In addition, implementation of small-scale HACCP systems can be suggested in shared kitchens to elevate consumer safety consciousness. It is also necessary to ease regulatory constraints through menu simplification and standardization, exemplified by practices in China. Furthermore, stimulating the growth of the sharing economy via business data-sharing platforms must be considered. Moreover, it is more practical and suitable to build a public business model rather than a personal business model to properly establish a shared kitchen in Korea.

Further research is imperative to undertake cost-effective assessment of the compulsory liability insurance applicable to domestic shared kitchens. Additionally, evaluating the feasibility of implementing small-scale HACCP systems to ensure the stable establishment of shared kitchens and to enhance food safety is also considered.

## Figures and Tables

**Figure 1 foods-13-00918-f001:**
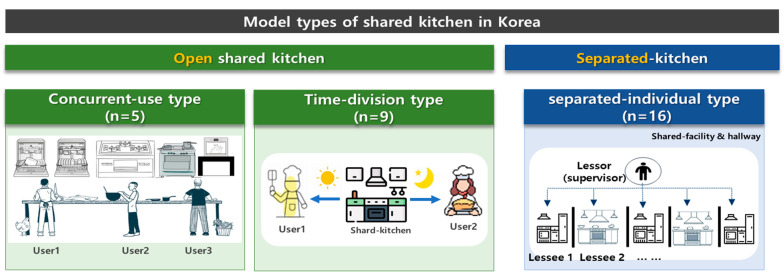
The classification of shared kitchen models in Korea.

**Table 1 foods-13-00918-t001:** HPLC gradient condition for detection of sorbic acid and parahydroxybenzoic acid.

Time (min)	Solvent A *	Solvent B **
0.0	75	25
2.5	75	25
7.0	65	35
12.0	60	40
15.0	70	30

* Solvent A: 1 L of distilled water containing 2.5 g of 40% tetrabutylammonium hydroxide and 1.2 g of 85% phosphoric acid. ** Solvent B: acetonitrile.

**Table 2 foods-13-00918-t002:** Microbial contamination levels of facilities and environment according to the types of shared kitchens.

Category	Microbial Hazards	Concurrent-Use	Time-Division	Separated-Individual
Hallway	*n* *	38	8	33
	TAC **	ND ***	ND	ND
	Coliforms **	ND	ND	ND
	Yeast and mold **	0.23 ± 0.29 ^a^	0.06 ± 0.17 ^a^	0.09 ± 0.24 ^a^
Refrigerator door	*n*	12	8	9
	TAC	0.85 ± 1.18 ^a^	1.94 ± 1.71 ^a^	1.45 ± 1.87 ^a^
	Coliforms	0.72 ± 1.30 ^a^	0.91 ± 1.28 ^a^	0.43 ± 1.28 ^a^
	Yeast and mold	ND	ND	ND
Refrigerator inside	*n*	18	6	15
	TAC	1.64 ± 1.68 ^ab^	2.31 ± 2.22 ^a^	3.90 ± 2.44 ^b^
	Coliforms	0.61 ± 1.34 ^a^	0.12 ± 0.29 ^a^	2.37 ± 2.17 ^b^
	Yeast and mold	ND	ND	ND
Sink and worktop	*n*	12	2	10
	TAC	1.48 ± 1.68 ^a^	4.48 ± 1.22 ^a^	4.31 ± 3.01 ^a^
	Coliforms	0.71 ± 1.11 ^a^	2.54 ± 3.59 ^a^	2.31 ± 2.64 ^a^
	Yeast and mold	ND	ND	ND
Cutlery	*n*	18	4	15
	TAC	0.88 ± 1.42 ^ab^	2.16 ± 1.80 ^a^	3.57 ± 2.21 ^b^
	Coliforms	0.42 ± 1.00 ^a^	0.97 ± 1.06 ^a^	1.80 ± 1.94 ^a^
	Yeast and mold	ND	ND	ND
Cooking clothes	*n*	5	2	5
	TAC	0.81 ± 0.85 ^a^	1.44 ± 0.06 ^a^	3.26 ± 2.09 ^a^
	Coliforms	ND	0.35 ± 0.49 ^a^	1.90 ± 2.00 ^a^
	Yeast and mold	ND	ND	ND

* *n*: numbers of samples, ** unit: CFU/100 cm^2^, *** ND: not detected, TAC: total aerobic bacteria. The values are presented as a mean ± standard deviation. ^a,b^ Means with different letters in each category are significantly different in the types of shared kitchens (*p* < 0.05).

**Table 3 foods-13-00918-t003:** The microbial hazard levels of various food cooked in the shared kitchens.

Microbial Hazard	Cheese Cake	Plain Scone	Peanut Butter Cookie	Chocolate Cookie	Quinoa Tofu Mushroom Salad	Shrimp Quinoa Salad	Sauce
Total aerobic bacteria	1.09 ± 0.43	0.70 ± 0.30	0.48 ± 0.17	0.42 ± 0.00	3.93 ± 0.24	3.27 ± 0.68	ND
Coliforms	ND	ND	ND	ND	1.42 ± 0.15	0.97 ± 0.60	ND

The values are presented as a mean ± standard deviation. ND: not detected.

**Table 4 foods-13-00918-t004:** The chemical hazard levels of various food cooked in the shared kitchens.

Category		Cheese Cake	Plain Scone	Peanut Butter Cookie	Chocolate Cookie	Quinoa Tofu Mushroom Salad	Shrimp Quinoa Salad	Sauce
Preservatives * (g/kg)		ND	-	-	-	-	-	ND
Total aflatoxin (μg/kg)		-	-	ND	-	-	-	-
Pesticide Residue (mg/kg)	Dinotefuran	-	-	-	-	0.033	0.016	-
	Imazalil	-	-	-	-	0.025	0.068	-
	Lufenuron	-	-	-	-	0.027	0.025	-
	Pyridalyl	-	-	-	-	0.054	0.052	-
Heavy metal (mg/kg)	Cadmium	-	-	-	-	0.0095	0.0033	-
	Lead	-	-	-	-	0.0042	0.0047	-
Tar color		-	-	-	-	-	-	ND

The values are presented as a mean ± standard deviation. ND: not detected, -: not tested. * Sorbic acid, ethyl p-hydroxybenzoate, methyl p-hydroxybenzoate, extra preservative, and total preservative were analyzed.

**Table 5 foods-13-00918-t005:** Compliance assessment for hygiene conditions of shared kitchen according to the type of shared kitchens.

Category	*n*	Score of Compliance		
		Concurrent-Use	Time-Division	Separated- Individual
Regulation	18	0.60 ± 0.55 ^a^	2.11 ± 2.09 ^ab^	2.81 ± 1.72 ^b^
Workstation and environment	47	12.00 ± 11.64 ^a^	15.00 ± 4.90 ^a^	15.38 ± 7.53 ^a^
Hygiene management	51	13.40 ± 9.40 ^a^	17.11 ± 11.60 ^a^	22.06 ± 6.35 ^a^
Food defense	2	0.40 ± 0.55 ^a^	2.00 ± 1.00 ^b^	2.06 ± 0.77 ^b^
Document management	10	2.20 ± 2.05 ^a^	1.89 ± 2.71 ^a^	5.25 ± 2.74 ^b^

*n*: Numbers of evaluation items in each category. The values are presented as a mean ± standard deviation. The higher the scores, the less compliance with the inspection items (0: compliance and 1: noncompliance). ^a,b^ Means with different letters in each category are significantly different in the types of shared kitchens (*p* < 0.05).

**Table 6 foods-13-00918-t006:** The responds of survey to shared kitchen operators.

Survey Question	Response	Concurrent-Use (*n* = 2)	Time-Division (*n* = 4)	Separated- Individual *(n* = 14)	Total
Q 26: Does the manager regularly clean and disinfect the shared facility for hygiene management? If yes, what is the frequency of cleaning and disinfection?	Not applicable	-	-	1 (7.1%) *	1 (5%) **
	Monthly	-	1 (25.0%)	6 (42.9%)	7 (35%)
	Biweekly	-	-	2 (14.3%)	2 (10%)
	Weekly	-	1 (25.0%)	3 (21.5%)	4 (20%)
	Thrice a week	1 (50.0%)	-	-	1 (5%)
	Daily	1 (50.0%)	2 (50.0%)	1 (7.1%)	4 (20%)
	Other	-	-	1 (7.1%)	1 (5%)
Q 29: Are separate hygiene inspections conducted in the shared kitchen? If yes, what is the frequency of these inspections?	Daily	1 (50.0%)	1 (25.0%)	-	2 (10%)
	More than three times a week	-	-	-	-
	Twice a week	-	-	-	-
	Once a week	1 (50.0%)	2 (50.0%)	2 (14.3%)	5 (25%)
	Once a month	-	-	7 (50.0%)	7 (35%)
	No separate management	-	1 (25.0%)	5 (35.7%)	6 (30%)
Q 33: Is there ongoing hygiene training for managers, including a designated hygiene manager, in the shared kitchen? If yes, please specify the frequency.	Not conducted	-	-	9 (64.3%)	9 (45%)
	More than three times a week	-	-	-	-
	Once a week	-	2 (50.0%)	-	2 (10%)
	Once every two weeks	-	-	-	-
	Once a month	1 (50.0%)	2 (50.0%)	4 (28.6%)	7 (35%)
	Once a year	1 (50.0%)	-	1 (7.1%)	2 (10%)
	Other (please specify)	-	-	-	-
Q 41: What are the major challenges in the shared kitchen you are currently managing?	Hygiene management by shared kitchen users (contractors)	-	1 (25.0%)	4 (28.6%)	5 (25%)
	Hygiene management of shared kitchen facilities	1 (50.0%)	1 (25.0%)	2 (14.3%)	4 (20%)
	Management of the inspection and selection of ingredients (entry and exit of food)	-	-	-	-
	Control of outsiders (including delivery personnel)	-	1 (25.0%)	3 (21.4%)	4 (20%)
	Costs associated with user contracts (rent, deposit, goodwill, etc.)	1 (50.0%)	1 (25.0%)	5 (35.7%)	7 (35%)
	Other (please specify)	-	-	-	-

* The results are presented as the percentage of each type of shared kitchen. ** The results are represented as the percent of total number of shared kitchens.

## Data Availability

The original contributions presented in the study are included in the article, further inquiries can be directed to the corresponding author.

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
