# Peer review of "Assessment of Hygiene Management Practices and Comparative Analysis of Regulatory Frameworks for Shared Kitchens across Different Countries"

_foods, 2024, doi:10.3390/foods13060918_

Round 1

Reviewer 1 Report

Comments and Suggestions for Authors

The authors describe interesting research about the new phenomenon, the shared kitchen.

The following revisions could be considered:

Line 48: "through regulatory sandboxes": can this be explained in more detailed. I did not catch the argument.

Line 57, 65: is contraction the correct word? Revise

Line 77: move the figure  1 nearer to where it is mentioned

Line 76, 81: avoid overlap in the methods section. The same things are said several times.

Line 94, 113 and throughout: add space between number and unit

Line 125: what is the rationale for selection of chemical hazards. Most of them are not related to kitchen hygienen or kitchen sharing, but rather from the food supply and agricultural practices.

Line 147-154, 192-195: avoid too much details in methods description, especially for standard methods. Rather add some validation results.

Line 227: can the review strategy be detailed her in more detail. The selection of countries and regions in the US sounds rather arbitrary.

Lines 234-237: this is an unnecessary repetiton of information already contained twice in the methods section. Avoid overlap and condense the already rather long text.

Lines 284, 287 and throughout: the different units for CFU lead to difficult comparisons between studies. Can this be aligned?

Table 2: units should be added

Lines 307, 331: please show the data in a data annex, supplements (check data requirements of MDPI)

3.2.2: The section should be revised focusing on chemical hazards with relevance to kitchen sharing only

Table 6: can the references for the information be provided?

Section 3.5.2: UK is not part of EU anymore, consider moving the UK information into a separate section

Lines 614-624: several mandatory sections of foods are missing, including data availability, re-check journal template

Author Response

Thank you 

Reviewer 2 Report

Comments and Suggestions for Authors

The paper represents an extremely interesting research that evaluates hygiene management practices and compares the regulatory frameworks for shared kitchens in different countries, however, the following needs to be supplemented and refined:

Materials and Methods

lack of clarification of what solvents A and B were in the HPLC method (and clearly connect them with the text and table 1)

Table 2, only for the classifications "Refrigerator inside" and "Cutlery" are added letters that are an indicator of similarities or statistically significant differences - however, it is not clear how to "read" the similarities and/or differences - if it is inside a line - then something with the association of letters is not OK, if it is inside the column, then in the others classification. groups are missing labels.

It is necessary to add marks of statistically significant similarity or difference, but for all tables and observed parameters

Table 3

the presentation of the data in table 3 is extremely confusing and I suggest separating those products for which pesticide residues have been determined

Table 4 

it is not clear what the letters next to some of the results represent, while some do not, and it is not clear what you meant by the text below the table "(0: compliance and 1: non-compliance)"

table 5 - it is not clear what the last column represents, for example, for Q26 and "Not applicable" there is 1 respondent available, which is 7.1%, and that is a total of 5% that corresponds to 1 respondent, and if you add up the penultimate column for Q26 - the sum is not 100%

Please check all the information in the table

Table 6

for the statements in table 6, it is mandatory to add references for the mentioned laws and regulations

Sincerely,

Author Response

Thank you 

Reviewer 3 Report

Comments and Suggestions for Authors

The paper entitled “Assessment of Hygiene Management Practices and Comparative Analysis of Regulatory Frameworks for Shared Kitchens Across Different Countries” authored by Yu Jin Na, Jin Young Baek, So Young Gwon and Ki Sun Yoon, deals with monitoring and assessment of microbial and chemical hazards in shared kitchens.

General comments:

-        The title is suitable and completely reflects the aims of the study.

-        The study is well designed, the aims and results are clearly presented.

-        The abstract briefly describes the purpose of the study and main findings.

Introduction:

-        I could not open several cited references in the form of web links (ref. 2, 3, 8, 30, 32, 34)

-        The references cited in Introduction should be more relevant in terms of scientific background. More scientific papers and previous studies should be cited, commented and analyzed here.

-        Last paragraph: Please, briefly mention the methods used for achieving the aims of the study.

Material and Methods:

-        Subsection 2.1: The mentioned Figure 1 should be placed close to this paragraph.

-        Most of the methods are well described, however the statistical part (subsection 2.8) should be more explained. Consider other methods for sample comparison, ANOVA is too basic.

Results and Discussion:

-        Nicely presented. Some additional statistics should be added in order to enhance the scientific soundness of the study.

-        Consider the results presentation in the form of graphs so they can be easily interpreted.

Conclusions:

-        Well written, but consider adding a paragraph containing the recommendations for further research regarding the topic.

Author Response

Thank you for your time 

Round 2

Reviewer 2 Report

Comments and Suggestions for Authors

Thank you for accepting my comments. the work is much clearer now.

The only thing that is still unclear to me as a reader is that in some lines you mark the significance of the differences with letters, and in the same table there are some lines without marks. Shown in this way, it seems as if it was "forgotten" because the values are obviously significantly different. If there are no differences, I suggest that you put the same letters, or that you explain below the table why some data do not have an additional associated letter.

Sincerely,

Reviewer 3 Report

Comments and Suggestions for Authors

The authors corrected the manuscript according to the suggestions. Now, I recommend its publication.